# Imaging the transient heat generation of individual nanostructures with a mechanoresponsive polymer

Xueqin Chen[1], Qing Xia[1], Yue Cao[1], Qianhao Min[1], Jianrong Zhang[1], Zixuan Chen[1], Hong-Yuan Chen[1] & Jun-Jie Zhu[1]

Measuring the localized transient heat generation is critical for developing applications of nanomaterials in areas of photothermal therapy (PTT), drug delivery, optomechanics and biological processes engineering. However, accurate thermometry with high spatiotemporal resolution is still a challenge. Here we develop a thermosensitive polymer-capped gold nanorod (AuNRs@pNIPAAm), which has temperature-dependent local surface plasmon resonance spectra due to the submolecular conformational change of pNIPAAm molecules. We measure the conformational dynamics on individual gold nanorods at the milliseconds level by the developed spatiotemporal resolution plasmonic spectroscopy (SRPS) and find that it has a fast (<4 ms), linear and reversible mechanoresponse to temperature changes as small as 80 mK. The rapid and highly sensitive thermosensitive AuNRs@pNIPAAm opens a new way to achieve spatiotemporal thermometry for potential applications in PTT and other biological research.

---

[1] State Key Laboratory of Analytical Chemistry for Life Science and Collaborative Innovation Center of Chemistry for Life Sciences, School of Chemistry and Chemical Engineering, Nanjing University, 163 Xianlin Ave, 210023 Nanjing, China. Xueqin Chen and Qing Xia contributed equally to this work. Correspondence and requests for materials should be addressed to Z.C. (email: chenzixuan@nju.edu.cn) or to J.-J.Z. (email: jjzhu@nju.edu.cn)

The capability to image transient heat generation of nanomaterials with high spatiotemporal resolution is critical for the development of applications in many areas, such as photothermal therapy (PTT)[1–3], drug delivery[4–6], opto-mechanics[7,8], and biological processes engineering[9,10]. In particular, accurate photothermal thermometry at the single-nanoparticle level provides a valuable tool for in situ cellular research. For instance, current PTT approaches generate local heat sources in tumor cells using photoabsorbing nanomaterials, but uncontrollable temperature may damage neighboring healthy cells. A temperature-feedback photothermal approach offers a solution by controlling the photothermal temperature with the real-time thermometry[11–13]. Numerous approaches have been developed to attain highly sensitive intracellular thermometry methods, including fluorescent probes[14–18] and nanodiamonds[11,19], but they are limited by spatiotemporal resolution. Spatial resolution could be improved by the encapsulation of photothermal agents with a thermometer[20] or scanning-based strategies[21,22], but the detection speed keeps the temporal resolution far from the aim of imaging the transient heat generation. Integration of the photothermal material (for example, gold nanorods) and thermosensitive polymers may be the solution due to distinct photothermal properties and the mechanoresponse. These types of polymers have a reversible conformational change in response to small temperature variations around a low critical solution temperature (LCST)[23–25], which leads to a rise in the adjacent refractive index of capped nanostructures.

Plasmonic resonance scattering of metal or semiconductor nanomaterials has attracted considerable attention due to its stable scattering, which is sensitive to the adjacent refractive index. Many efforts have been made to develop an advanced dark-field microscope (DFM) for plasmonic nanoprobes in chemical and biological systems, and the DFM technologies reported to date fall into two categories: spectroscopy[26–33] and scattering intensity[34–37] strategies. The former utilizes a broadband illuminator and spectrometer to analyze LSPR spectra, and the latter utilizes a single-color laser illuminator to describe the plasmonic scattering cross-section evolution. Each technology has its own unique merits and distinct drawbacks. For instance, spectroscopy strategies provide highly stable spectra capture for individual nanoparticles, but they suffer from low-throughput analysis and large biological scattering, which makes it difficult to detect nanoprobes inside cells. The scattering intensity strategy avoids the broadband scattering background from the biological system, and demonstrates extreme temporal resolution and sensitivity. However, power fluctuation and orientation-dependent scattering intensity makes it difficult to obtain the exact LSPR wavelength, thus limiting the potential application of this strategy in a complex biological system. A strategy combining the benefits of spectroscopy strategies in measurement stability and those of scattering intensity strategies in high spatiotemporal resolution imaging is still lacking.

Here we develop a thermosensitive polymer poly(N-isopropylacrylamide) capped gold nanorod (AuNRs@pNIPAAm) as the thermometer for high sensitivity imaging of transient heat generation. AuNRs@pNIPAAm has a temperature-dependent LSPR spectral shift ($\Delta\lambda_{peak}$) around LCST, without interference from ionic strength and pH values. After optimization, a LSPR spectral shift of more than 10 nm is obtained in response to temperature variation. To improve the scattering spectra measurement efficiency for large scale AuNRs@pNIPAAm samples, we develop an original spatiotemporal resolution plasmonic spectroscopy (SRPS) strategy based on a homemade two-channel highly inclined and laminated optical (HILO)[38,39] sheet set-up. With SRPS, we can simultaneously and directly measure the temperature-dependent $\Delta\lambda_{peak}$ of multiple AuNRs@pNIPAAm

particles at the millisecond level. Extracellular and intracellular experimental results indicate that AuNRs@pNIPAAm has a linear response to temperature near LCST, and the response time is

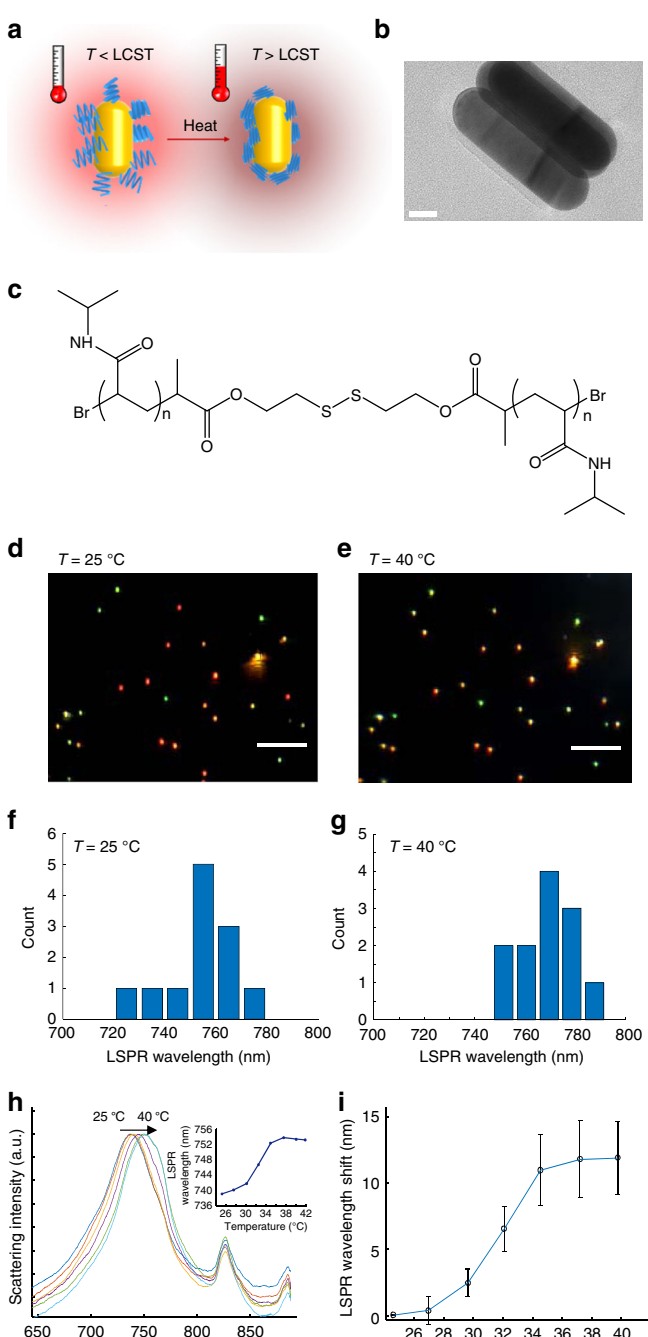

**Fig. 1** Schematic illustration of the thermometry principle of AuNR@pNIPAAm. **a** The temperature response mechanism of AuNR@pNIPAAm. **b** The high-resolution transmission electron microscopy image of AuNR@pNIPAAm. Scale bar is 20 nm. **c** The molecular structure of pNIPAAm for AuNR@pNIPAAm fabrication. **d** A broadband DFM image of AuNR@pNIPAAm at 25 °C. Scale bar is 5 μm. **e** Same as **d** but the temperature is increased to 40 °C. **f** The LSPR wavelength distribution of AuNR@pNIPAAm at 25 °C. **g** Same as **f** but the temperature is increased to 40 °C. **h** Scattering spectra of a single AuNR@pNIPAAm particle over the temperature. Inset: Temperature-dependent LSPR wavelength of this particle. **i** Mean scattering spectral redshifts of single AuNRs@pNIPAAm particles as functions of the temperature. Error bars, ±1 s.d. with $n = 10$ each

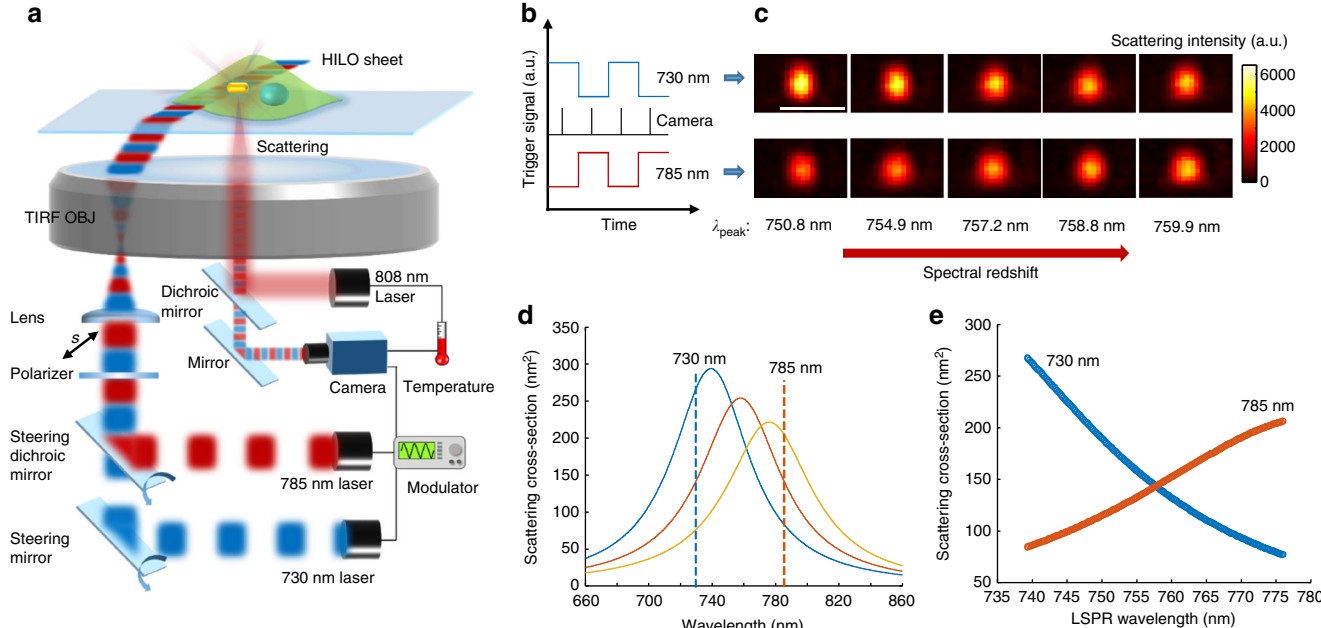

**Fig. 2** Work principle of the spatiotemporal resolution plasmonic spectroscopy (SRPS). **a** Schematic illustration of SRPS set-up. An s-polarized two-color pulse beam with switchable wavelength between 730 and 785 nm is generated by combining two antiphase pulse lasers, which is then directed onto the cover slide with a high incident angle to form the highly inclined and laminated optical (HILO) sheet. The thin optical sheet excites the LSPR scattering of gold nanorods. The scattering light is then collected by a camera to form two-channel dark-field images, which are then converted to the spectral mapping. Another 808-nm laser is focused on the single-nanoparticle for in situ heat generation. **b** Trigger signal waveforms of the 730-nm (blue), 785-nm (red) laser and the camera (black) are generated by a waveform generator. **c** Two-channel snapshots of a single gold nanorod with the LSPR wavelength adjusted from 750.8 to 759.9 nm. **d** Simulated scattering spectra of a gold nanorod when adjacent dielectric functions are changed from 1.67 to 1.87. **e** Scattering cross-sections of the gold nanorod in **d** at 730 (blue) and 785 nm (red) as functions of the LSPR wavelength

estimated to be less than 4 ms. The proposed strategy enables us to achieve spatiotemporal intracellular thermometry for potential applications in PTT and provides a valuable tool for widespread biological research.

## Results

**Design of AuNRs@pNIPAAm probes for thermometry.** As shown in Fig. 1a, the thermosensitive polymer pNIPAAm is covalently bonded to the gold nanorod surface via the Au-S bond (Fig. 1c) and has a stretch conformation at room temperature. The assembled pNIPAAm layer has a uniform thickness of ~6 nm in the dry state, which is determined by high-resolution transmission electron microscopy (Fig. 1b). When the temperature increases over LCST, stretched pNIPAAm collapses (Supplementary Fig. 1), inducing the increment of the adjacent refractive index and the spectral redshift $\Delta\lambda_{peak}$ of the gold nanorod. Initially, we investigate the $\Delta\lambda_{peak}$ response of individual AuNRs@pNIPAAm particles to a static state temperature. Figure 1d demonstrates the broadband DFM image of abundant individual AuNRs@pNIPAAm particles deposited on a glass slide while the solution temperature is maintained at 25 °C. Clear red dots and the scattering spectra indicate a uniform LSPR wavelength distribution around 750 nm (Fig. 1f). In contrast, when the temperature increases to 40 °C, these red scattering dots fade (Fig. 1e) because the LSPR wavelength grows near the infrared region (Fig. 1g).

We choose a single AuNRs@pNIPAAm particle to demonstrate the temperature-dependent $\Delta\lambda_{peak}$ (Fig. 1h). Scattering spectra are observed as shifting from 738 to 751 nm when the temperature increases from 25 to 40 °C. Notice that the two peaks located at 820 and 880 nm are assigned to the intrinsic spectrum of the white light source. The s-shape temperature-dependent $\Delta\lambda_{peak}$ curve illustrates a linear range from 30 to 35 °C, closely

matching the LCST (32 °C) of pNIPAAm (Fig. 1h, inset). To evaluate the variability of different particles in temperature response, the mean $\Delta\lambda_{peak}$ of a large number of particles is plotted by the temperature (Fig. 1i). A large deviation demonstrates the significance of single-nanoparticle calibration. For each AuNR@pNIPAAm particle, we can express the temperature ($T$) in range from 30 to 35 °C:

$$T = k_T \lambda_{peak} + b, \qquad (1)$$

where $k_T$ is the slope in the linear range and $b$ is a constant. Eq. (1) predicts that AuNRs@pNIPAAm allows single-nanoparticle thermometry. The thermometry range could be extended by integrating acrylamide (AAm) into the pNIPAAm because PTT control requires thermal readings between 35 °C and at least 50 °C. As shown in Supplementary Fig. 2, the LSPR wavelength of gold nanorods modified by the copolymer pNIPAAm-co-pAAm with LCST at 39 °C illustrates a two-stage s-shape curve over the temperature. Two linear ranges are located at 30–35 °C and 42–48.5 °C. The first linear range reproduces the pure pNIPAAm capped gold nanorods, while the second range is attributed to the integration of AAm. The extended thermometry range offers the application potential in PTT control.

**High-throughput spectral mapping of individual AuNRs@pNIPAAm.** To measure the dynamic temperature evolution, the broadband DFM is inefficient due to the low-throughput spectra capture ability. To obtain a rapid LSPR scattering spectral mapping of substantial AuNRs@pNIPAAm particles, we propose and demonstrate SRPS. As illustrated in Fig. 2a, SRPS is performed on a homemade two-channel total internal reflection (TIR) microscope equipped with a high numerical aperture oil immersion objective. Gold nanorods with a LSPR wavelength of around 750 nm are deposited on the cover slide and alternatively

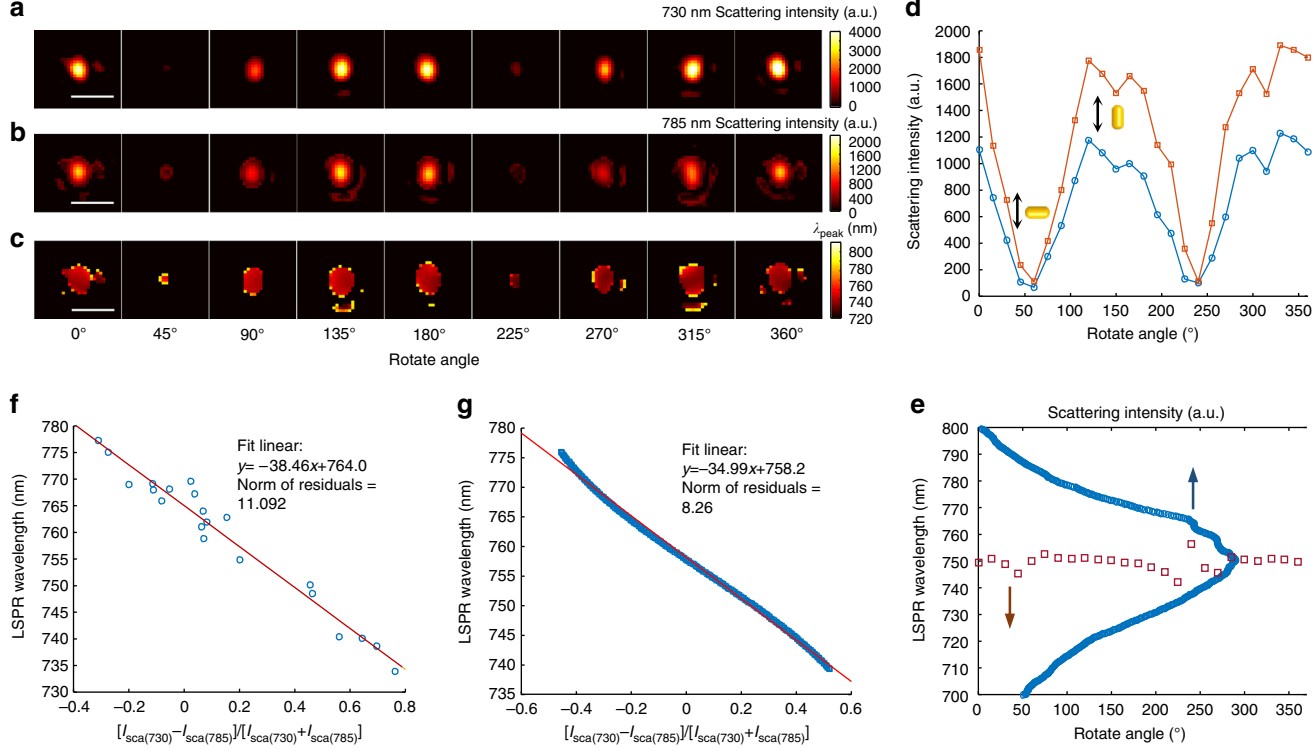

**Fig. 3** Accuracy and calibration of the spatiotemporal resolution plasmonic spectroscopy (SRPS). **a–c** 730 nm channel **a**, 785 nm channel **b** dark-field snapshots and converted LSPR wavelength images **c** of a single gold nanorod which rotated from 0 to 360°. Scale bar is 1 μm. **d** Rotation angle-dependent scattering intensity of 730-nm (squares) and 785-nm (circles) channel snapshots. **e** Rotation angle-dependent LSPR wavelength (squares) and scattering spectra obtained by a commercial spectrograph (circles) of the same gold nanorod. **f** A calibration curve illustrating the linear relationship between values of $[I_{\mathrm{sca}(730)} - I_{\mathrm{sca}(785)}]/[I_{\mathrm{sca}(730)} + I_{\mathrm{sca}(785)}]$ and scattering spectral peak wavelengths of single gold nanorods. **g** A calibration curve illustrating the linear relationship between simulated values of $[I_{\mathrm{sca}(730)} - I_{\mathrm{sca}(785)}]/[I_{\mathrm{sca}(730)} + I_{\mathrm{sca}(785)}]$ and simulated LSPR wavelengths

incubated with cells. Two s-polarized continuous lasers (with wavelengths of 730 and 785 nm) are modulated to generate two antiphase pulse beams (Fig. 2b). The two pulse beams are combined to generate a two-color pulse beam, whose wavelength periodically swaps between 730 and 785 nm. This two-color pulse beam is then directed onto the cover slide with an appropriate incident angle (slightly below the TIR critical angle) via the objective to generate a HILO sheet[38,39]. The thin optical sheet, generally less than tens of micrometers, is suitable for high spatial resolution imaging of nanoparticles inside cell samples. The scattering light of gold nanorods is collected by the same objective to form dark-field images, which are recorded by a complementary metal oxide semiconductor (CMOS) camera. To obtain the LSPR scattering wavelength of gold nanorods, the CMOS camera is synchronized with the two-color pulse beam to periodically capture dark-field images from the two wavelength lasers (Fig. 2b), generating a two-channel image sequence. We compare the two-channel snapshots of a single gold nanorod whose LSPR wavelength is adjusted from 750.8 to 759.9 nm (Fig. 2c). Initially, the gold nanorod has a stronger scattering at 730 nm. With the LSPR wavelength growing, the scattering at 730 nm becomes weaker, whereas the scattering at 785 nm becomes stronger. This result is explained by Mie's theory (details in Methods section). In short, Fig. 2d displays a series of simulated scattering spectra of gold nanorods, whose LSPR wavelengths are tuned from 740 to 775 nm by changing the adjacent dielectric functions from 1.67 to 1.87. These spectra have similar shapes, and the scattering cross-sections at 730 ($C_{730}$) and 785 ($C_{785}$) nm are always located on opposite sides of the peak of each curve. We calculate and plot $C_{730}$ and $C_{785}$ as functions of the LSPR wavelength in Fig. 2e, and two inverse curves are displayed.

Both curves demonstrate an approximately linear relation between the scattering cross-section and the LSPR wavelength, with a pair of opposite slopes, reproducing the experimental result displayed in Fig. 2c.

To convert these two-channel snapshots to spectral mapping, we solved the LSPR wavelength-dependent scattering cross-section equations at 730 and 785 nm (details in Methods section). Herein, $\lambda_{\mathrm{peak}}$ is expressed:

$$\lambda_{\mathrm{peak}} = -A\frac{I_{\mathrm{sca}(730)} - I_{\mathrm{sca}(785)}}{I_{\mathrm{sca}(730)} + I_{\mathrm{sca}(785)}} + \lambda_0, \qquad (2)$$

where $A$ and $\lambda_0$ are constants, and $I_{\mathrm{sca}}$ is the scattering intensity of each channel. Eq. (2) predicts that SRPS provides a new way to measure $\lambda_{\mathrm{peak}}$ without interference from direction dynamics of anisotropic nanoparticles, which is inevitable during optical intensity-based detection. To confirm, we perform the following rotation experiment. Figure 3a–c present two-channel snapshots and the corresponding spectral mapping snapshots of a single gold nanorod, rotating from 0 to 360°. The scattering intensity of both channels illustrates a periodic fluctuation, but the obtained spectral mapping remains at a stable value (see also in Supplementary Movie 1–3). Figure 3d plots a rotated angle $\theta$-dependent intensity in each channel, indicating that $I_{\mathrm{sca}} \propto \cos^2\theta$. In contrast, $\lambda_{\mathrm{peak}}$ remains steady at around 750 nm in most orientations, with the exception of when the polarization of incident laser is orthogonal to the long axis of the gold nanorod (Fig. 3e, squares). In this case, the LSPR scattering of gold nanorods is significantly reduced, and the transverse surface plasmon resonance (TSPR) scattering dominates[40]. Notably, the calculated $\lambda_{\mathrm{peak}}$ dramatically reproduces the scattering spectrum of the target gold nanorod captured by a commercial

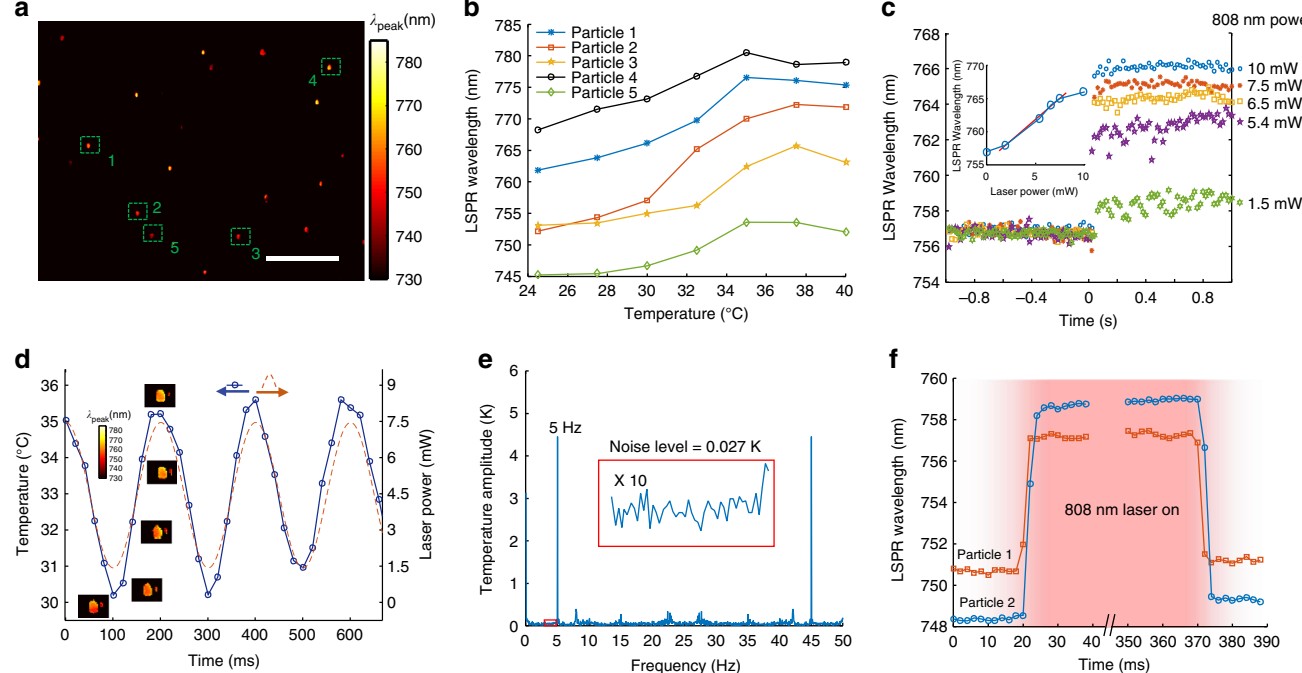

**Fig. 4** Spatiotemporal thermometry by the spatiotemporal resolution plasmonic spectroscopy (SRPS). **a** LSPR spectral mapping of AuNRs@pNIPAAm at 25 °C. Scale bar is 10 μm. **b** Temperature-dependent scattering spectral peak wavelengths of selected AuNRs@pNIPAAm particles marked by squares in **a**. **c** LSPR wavelength of an AuNR@pNIPAAm particle numbered by 2 in **a** response to the 808-nm laser with power of 1.5 (hexagrams), 5.4 (pentagrams), 6.5 (squares), 7.5 (asterisks), and 10 mW (circles) at $t = 0$ s. In the inset, LSPR wavelength is plotted to the 808-nm laser power (circles) while a solid line is fitted using data points from 5.4 to 7.5 mW. **d** Time-dependent applied 808 nm laser power (dash line) and the obtained temperature of the same particle in **c** (circles). Insets, LSPR wavelength images of the heated AuNR@pNIPAAm particle producing the temperature data. **e** Fourier spectrum of the temperature data in **d**. Inset is 10 times magnification of the region marked by the solid square. **f** Time-dependent LSPR wavelengths of two AuNR@pNIPAAm particles response to the 808-nm laser on and off. Sample interval is 2 ms

spectrometer (Fig. 3e, circles), revealing that SRPS is a reliable spectrometric method, in consideration of the anisotropic nanoparticle orientation diversity in a biological system. To demonstrate the accuracy of SRPS, we perform the following calibration experiment. Gold nanorods are immersed in water, followed by adding a 25% sodium chloride solution to generate abundant gold nanorod samples with $\lambda_{peak}$ in the range of 740 −775 nm. We use SRPS and a commercial spectrometer to acquire values of $\frac{I_{sca(730)} - I_{sca(785)}}{I_{sca(730)} + I_{sca(785)}}$ and $\lambda_{peak}$ of each gold nanorod sample, respectively. As plotted in Fig. 3f, a well-fitted linear relationship, including the slope and intercept, follows the simulation result (Fig. 3g), revealing the high accuracy of SRPS in spectra measurement.

**Single-nanoparticle thermometry using SRPS**. The ability of SRPS in high-throughput LSPR spectral mapping provides us with the possibility of single-nanoparticle thermometry. To verify the thermo-response ability, we use SRPS to measure the temperature-dependent $\Delta\lambda_{peak}$ of individual AuNRs@pNIPAAm particles. Figure 4a presents the spectral mapping at 25 °C, and most of the particles demonstrate a LSPR wavelength in the range of 740−770 nm. We select five AuNRs@pNIPAAm particles to simultaneously investigate the $\Delta\lambda_{peak}$ in response to temperature variation, and the particles are marked by dash boxes. Figure 4b displays 5 s-shape temperature-dependent $\Delta\lambda_{peak}$ curves, similar to the results obtained by the spectrometer (Fig. 1h, inset).

To demonstrate the ability to detect transient heat generation, we use an 808-nm laser (Fig. 2a) to heat a single AuNR@pNI-PAAm particle (Fig. 4a, particle 2), whose $k_T$ and $b$ in eq. (1) are evaluated to be 0.41 K nm$^{-1}$ and −280 K, respectively (Fig. 4b, curve 2). The photothermal temperature of nanoparticle $T(r)$ is

calculated by

$$T(\mathrm{r}) = T_m + \frac{P_{abs}}{4\pi\kappa r}, \quad (3)$$

where $T_m$ is the environment temperature, $P_{abs}$ is the laser power absorbed by the nanoparticle, and $\kappa$ is the thermal conductivity of the surrounding liquid. Eq. (3) predicts that the photothermal temperature of AuNR@pNIPAAm is proportional to the 808-nm laser power. Thus, we adjust the 808-nm laser power to generate local heat at the target AuNR@pNIPAAm particle, whose $\lambda_{peak}$ is recorded in the interval of 20 ms (Fig. 4c). The time-dependent $\lambda_{peak}$ illustrates a steady baseline at around 756 nm until the 808-nm laser illumination is applied, leading a sudden rise of $\lambda_{peak}$. With the greater power of 808-nm laser, comes the greater increase of $\lambda_{peak}$. The photothermal-induced $\lambda_{peak}$ is plotted as a function of the 808-nm laser power (Fig. 4c, inset), fitting the calibration result (Fig. 4b, curve 2). Thereby the transient heat generation at this AuNR@pNIPAAm particle is well controlled by an 808-nm laser power in the range of 1.5 −7.5 mW, and eq. (3) is simplified to be

$$T = T_m + kP(^{\circ}C), \quad (4)$$

where $T_m$ is 28.5 °C, $k$ is 0.83 °C mW$^{-1}$, and $P$ is the incident 808 laser power.

We create a sinusoidal periodic oscillation in the 808-nm laser power (1.5–7.5 mW, 5 Hz) to illustrate the ultrafast thermometry of AuNR@pNIPAAm. Figure 4d demonstrates the temperature oscillation (solid line), overlapping the heating laser power modulation (dash line) and revealing the high temporal resolution (20 ms) thermometry. The accuracy of the thermometry system is also examined. We obtain the temperature modulation of this AuNR@pNIPAAm particle by performing fast

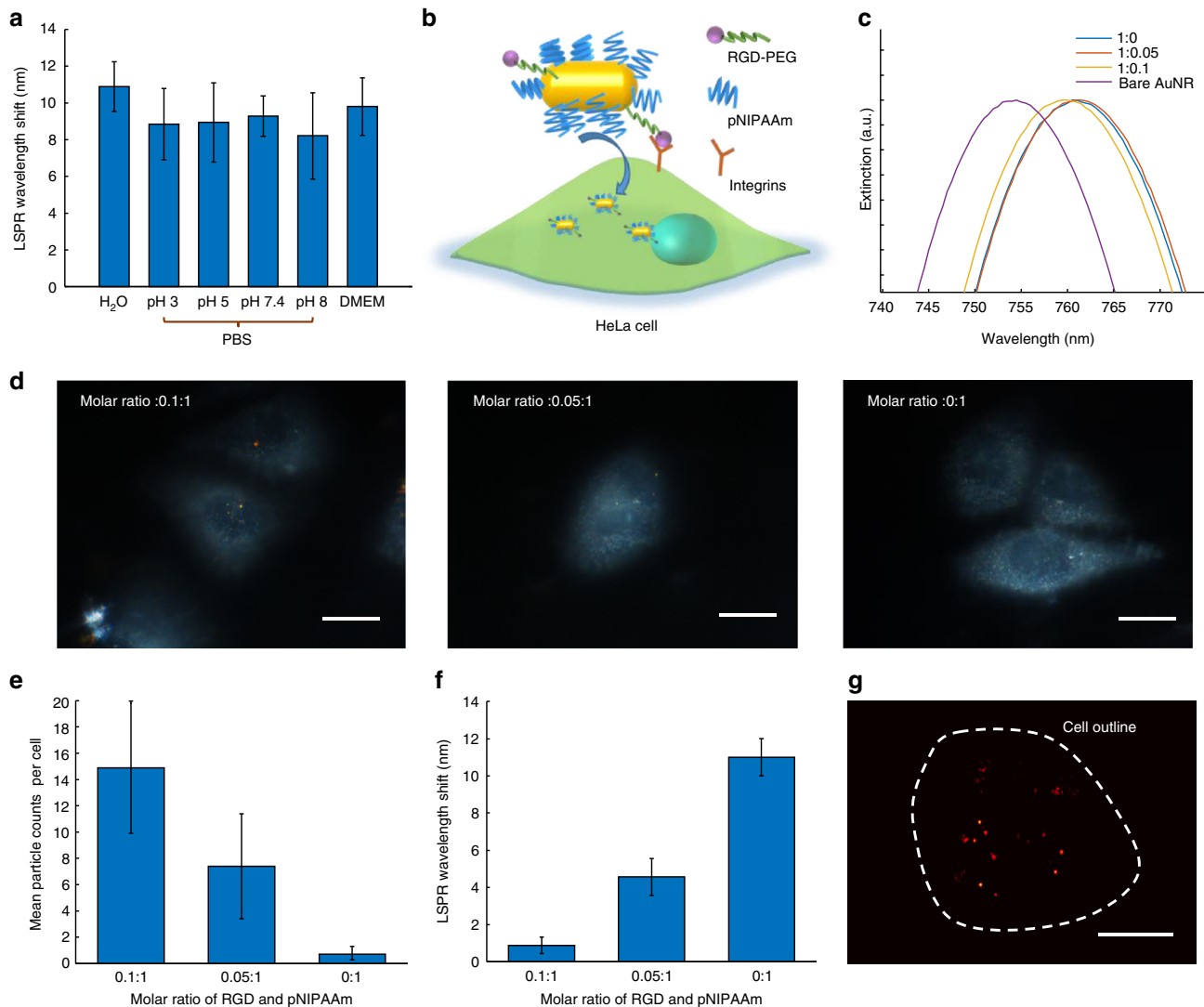

**Fig. 5** Optimization of molar ratio between Arg-Gly-Asp (RGD) peptides and pNIPAAm during AuNRs@pNIPAAm/RGD assembly. **a** LSPR wavelength shifts of AuNRs@pNIPAAm in water, phosphate buffered solutions (PBS) with pH from 3 to 8 and the cell culture media when the temperature increases from 25 to 40 °C. Error bars, ±1 s.d. with $n = 20$ each. **b** Schematic illustration of the HeLa cell-targeting AuNRs@pNIPAAm/RGD. **c** UV−vis spectra of bare gold nanorods and AuNRs@pNIPAAm/RGD assembled in mixture of RGD and pNIPAAm with molar ratio of 0.1:1, 0.05:1, and 0:1. **d** Broadband dark-field images of HeLa cells incubated with 40 pM AuNRs@pNIPAAm/RGD for 5 h. Scale bar is 20 μm. **e** Mean particle counts in a cell in the same sample shown in **d**. Error bars, ±1 s.d. with $n = 20$ each. **f** Temperature-induced LSPR wavelength shifts of AuNRs@pNIPAAm/RGD particles in **d**. Error bars, ±1 s.d. with $n = 10$ each. **g** The scattering image of AuNRs@pNIPAAm/RGD in a HeLa cell by the HILO sheet dark-field microscope. Scale bar is 10 μm

Fourier transform (FFT) analysis on the time-temperature curve in Fig. 4d. The FFT spectrum illustrates a peak located at 5 Hz (Fig. 4e). The peak amplitude is 4.5 K, with a noise level of 0.027 K, revealing a detection limit of 0.08 K (three times the noise level). The system noise is determined by mechanical drift, light source stability, and ambient temperature fluctuation. To investigate the ultimate temporal resolution, we reduce the sample interval to 2 ms by increasing the camera frame rate to 1000 fps. The time-elapse $\lambda_{peak}$ curves of two single AuNRs@pNIPAAm particles both show rapid responses (less than 4 ms) to turning the 808-nm laser on and off (Fig. 4f), attributed to the ultrafast mechanoresponse of pNIPAAm molecules on the gold nanorod surface.

**Single-nanoparticle thermometry in cancer cells**. Complex biological environments may introduce interference during intracellular thermometry, so we investigate the thermal response

of 20 AuNRs@pNIPAAm particles in different solutions, which could represent a variety of the whole samples. Figure 5a demonstrates the mean $\Delta\lambda_{peak}$ of the AuNRs@pNIPAAm particles immersed in water, phosphate buffered solutions (PBS) and the Dulbecco's modified eagle medium (DMEM) when the temperature increases from 25 to 40 °C. AuNRs@pNIPAAm maintains the thermometry ability in a broad pH range of solutions and the culture media though the ionic strength and biomolecules still affect sensitivity. For specific cell-targeting, Arg-Gly-Asp (RGD) peptides are conjugated to the gold nanorod surface together with pNIPAAm (AuNRs@pNIPAAm/RGD), which are specifically captured by integrins on cancer cell membrane Fig. 5b[41,42]. In our previous work, we found that RGD peptide coverage affects the response property of nanoprobes[42]. Thus, the mixed molar ratio of RGD peptides and pNIPAAm, which are then incubated with gold nanorods, is optimized to obtain the lowest ratio that still enables the specific cell-targeting. We prepare a series of AuNRs@pNIPAAm/RGD samples by

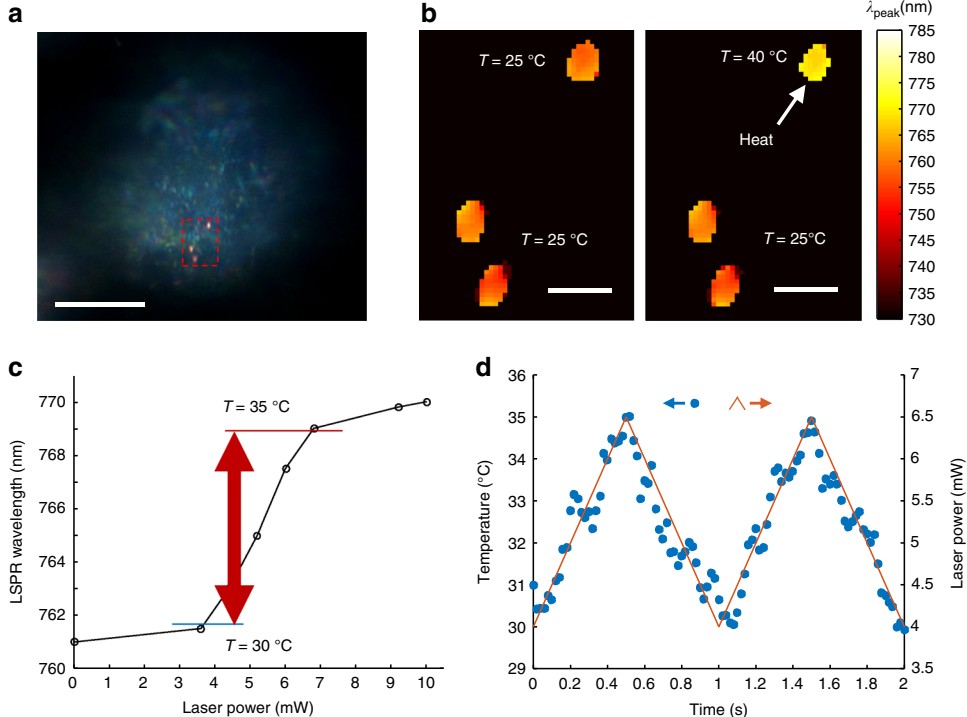

**Fig. 6** Illustration of the intracellular thermometry by AuNRs@pNIPAAm/RGD. **a** Broadband dark-field image of AuNRs@pNIPAAm/RGD particles inside a HeLa cell. The objective focuses on the middle height of the cell to image intracellular particles. Scale bar is 10 μm. **b** Spectral mapping of particles with or without the 808-nm laser illumination in the region marked by a dash square in **a**. Scale bar is 1 μm. **c** LSPR wavelength of the heated particle in **b** as a function of the 808-nm laser power. **d** Time-dependent applied 808 nm laser power (solid line) and the obtained temperature of the same particle in **c** (circles)

incubating the gold nanorods with mixtures of pNIPAAm and RGD peptides. The HeLa cell line is selected as the model to study the specific cell-targeting, in consideration of the abundant integrins on the membrane. As shown in Fig. 5d, clear scattering dots indicated that the individual AuNRs@pNIPAAm/RGD particles are inside the cells. The higher the molar ratio of RGD peptides and pNIPAAm, the more AuNRs@pNIPAAm/RGD particles are observed in HeLa cells after 5 h incubation, which is confirmed by the statistic result (Fig. 5e). However, the thermal response has an inverse trend due to the active sites occupation by RGD peptides (Fig. 5f). Thus the molar ratio of 0.05:1 is chosen for AuNRs@pNIPAAm/RGD preparation. In this case, AuNRs@pNIPAAm/RGD still demonstrates a reduced thermal response compared with AuNRs@pNIPAAm.

Commonly, photothermal agents require a higher concentration to generate sufficient heat to kill tumor cells, but highly dense particles will induce aggregation and cytotoxicity. Thus, the concentration of intracellular AuNRs@pNIPAAm/RGD needs to be optimized. As illustrated in Supplementary Fig. 3a, aggregates start to occur in HeLa cells incubated with 40 pM AuNRs@pNIPAAm/RGD for 5 h. These aggregates have broad scattering peaks, attributed to the scattering overlapping of multiple single particles (Supplementary Fig. 3b). After incubation for 8 h, massive aggregates are observed with much wider scattering peaks. The presence of broad scattering spectra may reduce the sensitivity, and a shorter incubation period less than 5 h may limit these. In this condition, AuNRs@pNIPAAm/RGD still demonstrates a good PTT performance for HeLa cells (Supplementary Fig. 4). Cytotoxicity is another essential factor of concern for foreign materials in a biological system. With the cell viability assay, HeLa cells displayed no viability variations, even after 48 h of incubation with 80 pM AuNRs@pNIPAAm/RGD, which is twice as much as the actual dosage (Supplementary Fig. 5).

As mentioned above, broadband scattering background of organelles prevents us from elucidating internal details. In contrast, the HILO dark-field microscope gives a clear image of more scattering dots (Fig. 5g), which is attributed to the thin incident laser sheet and low background scattering of the infrared laser. Furthermore, the HILO dark-field microscope enables us to reconstruct the three-dimensional (3D) scattering images of AuNRs@pNIPAAm/RGD inside the cell from z-scanned serial images (Supplementary Fig. 6, Supplementary Movie 4). Most AuNRs@pNIPAAm/RGD particles enter the cell and are evenly dispersed. This 3D image can be used to describe the spatial thermal distribution inside cells.

To demonstrate that single-nanoparticle thermometry is valid for the intracellular environment, we choose an AuNR@pNIPAAm/RGD particle inside the cell and heat it with the 808-nm laser to illustrate intracellular thermometry (Fig. 6a). We compare the target particle with two others in the same focal plane. As illustrated in Fig. 6b, the heated particle demonstrates a considerable response, while others have no apparent change, which is attributed to the high temperature gradient around the nanoscale heat source. To efficiently evaluate the local temperature, we use a simpler process instead of calibration for each particle. Simply, the AuNR@pNIPAAm/RGD particle demonstrates a $\Delta\lambda_{peak}$ as a function of the applied 808-nm laser power (Fig. 6c), fitting the temperature-dependent $\lambda_{peak}$ curves (Fig. 4b). Thus, the linear range from 762 to 769 nm indicates the temperature range of 30–35 °C, and the temperature is easily measured in this range. As shown in Fig. 6d, we generate a periodic triangle wave oscillation (4–6.5 mW, 1 Hz) to the applied laser power (solid curve) to illustrate the transient thermometry ability of the AuNR@pNIPAAm/RGD particle. We measure the real-time $\lambda_{peak}$ and evaluate the corresponding temperature, which follows the laser power extremely well

(circles). This proof-of-principle experiment indicates that AuNR@pNIPAAm/RGD is exceedingly suitable for the purpose of transient heat generation in a biological environment. Occasionally, we can observe the light-induced rotation of AuNR@pNIPAAm/RGD particles, which is more noticeable in those with mild thermal response. When heated by the 808-nm laser, the scattering intensity of an AuNR@pNIPAAm/RGD particle decreases in both channels (Supplementary Fig. 7a). In remarkable contrast, the converted $\lambda_{peak}$ remains steady at ~766 nm (Supplementary Fig. 7b). This direction-free spectral mapping by SRPS provides more accuracy in a biological process detection compared with the single-color laser DFM.

## Discussion

We have demonstrated a single-nanoparticle thermometer AuNR@pNIPAAm/RGD for imaging the transient heat genera-tion in cancer cells. The temperature-dependent LSPR wavelength of AuNR@pNIPAAm/RGD was measured in a high-throughput way by the developed SRPS, which combines the spectra recording stability merit of broadband DFM and spatiotemporal resolution of laser DFM (submicrometer and millisecond level). The temporal resolution of the SRPS can be further enhanced by increasing the pulse beam frequency and camera frame rate, which may enable observations of transient state products during chemical reactions. Optimizing the preparation of AuNRs@p-NIPAAm enables us to accurately measure 80 mK of temperature change in solution. The uptake of AuNRs@pNIPAAm/RGD can be used to demonstrate the 3D thermal distribution inside cancer cells, which offers many possibilities. For example, integration of acrylamide into pNIPAAm enables tuning of LCST across a wide range of temperatures[25], which offers real-time observations of dynamic subcellular processes in biological and medical applications[11]. Furthermore, the endoscope system equipped with the SRPS may enable the monitoring of plasmonic nanoprobes in deep tissue or cells, which will expand potential applications in vivo. Our experimental results reveal that only the positions where photothermal agents are located can generate sufficient heat during a PTT process. Thus, an effective cure process results in the concentration of abundant photothermal agents in target area, and this process prevents material consumption, which is dangerous for healthy neighboring cells. SRPS provides a possible solution for this, considering its capability in both photothermal properties and spatiotemporal thermometry. By targeting photothermal agents to vital cells or organelles in tumor tissues, it is possible to heat and detect the temperature of those located areas accurately. This precision guided nanoheaters will destroy the target cell or tissue, while keeping the neighboring healthy cells safe.

## Methods

**Spatiotemporal resolution plasmonic spectroscopy set-up.** As illustrated in Fig. 2a, SRPS was performed on a homemade two-channel HILO sheet dark-field microscope, which was established based on a Nikon Ti-E inverted microscope with three independent total internal reflection (TIR) illumination paths for a 730-nm laser (HL7302MG, Thorlabs), a 785-nm laser (LP785-SF100, Thorlabs) and a broadband light source (EQ-99XFC LDLS, Energetiq Technology). The experiment used a Nikon TIRF ×100, oil immersion, NA = 1.49 objective for HILO sheet generation and scattering light collection. A CMOS camera (Pike F-032B, Allied Vision) was used for two-channel dark-field imaging. An 808-nm (M9-808-0150, Thorlabs) laser was used as the heating light source. A dichroic mirror (FF791-SDi01, Semrock) was used to prevent the 808-nm laser from entering the camera. A dichroic mirror (FF740-Di01, Semrock) was used to combine the 730-nm and 785-nm lasers. Other optical elements and optomechanical compo-nents were all purchased from Thorlabs, Inc. Waveform generators (Keysight 33612A, Agilent) were used to build a two-color pulse laser (730/785 nm) and trigger the camera. Three trigger signals were synchronized by an oscilloscope (UPO2104CS, LINI-T). All device control and data processing were performed with Matlab software.

**Converting the two-color imaging to spectral mapping.** We use a spectrum model[43] to demonstrate how to convert the two-channel dark-field image to $\lambda_{peak}$ image. Figure 2c presents two-channel snapshots of a single gold nanorod immersed in a series of refractive index solution. Initially, the 730-nm channel has stronger scattering than that of 785-nm channel. Increasing the adjacent refractive index, the 730-nm channel scattering becomes weaker, whereas a growing 785-nm channel scattering intensity is observed. The scattering cross-section $C_{sca}$ of individual gold nanorods is expressed by Mie's theory[43]

$$C_{sca} = \frac{k^4}{6\pi}|\alpha|^2,$$ (5)

$$k = 2\pi/\lambda,$$

where $\alpha$ is the polarizability $k = 2\pi/\lambda$ and $\lambda$ is the incident light wavelength. Figure 2d displays a series of simulated scattering spectra of gold nanorods whose LSPR wavelength is tuned from 740 to 775 nm by changing the adjacent dielectric functions from 1.67 to 1.87. These spectra have similar shapes, and the scattering cross-sections at 730 ($C_{730}$) and 785 ($C_{785}$) nm are always located on opposite sides of the peak of each curve. We calculate and plot $C_{730}$ and $C_{785}$ as functions of LSPR wavelength in Fig. 2e, and an inverse variation trend is witnessed. Both curves demonstrate an approximately linear relation between the scattering cross-section and the LSPR wavelength with a pair of opposite slopes, reproducing the result displayed in Fig. 2c. In terms of this, $C_{730}$ and $C_{785}$ are approximately expressed by:

$$C_{730} = k(\lambda_0 - \lambda_{peak}) + C_0,$$ (6)

$$C_{785} = -k(\lambda_0 - \lambda_{peak}) + C_0,$$ (7)

where $k$ is the absolute value of curve slopes, $\lambda_{peak}$ is LSPR wavelength, $\lambda_0$ is the wavelength where two curves cross, and $C_0$ is the scattering cross-section at this cross point. Herein, $\lambda_{peak}$ is calculated by solving eq. (6), (7):

$$\lambda_{peak} = -\frac{C_0}{k}\frac{C_{730} - C_{785}}{C_{730} + C_{785}} + \lambda_0.$$ (8)

Eq. (8) predicts that $\lambda_{peak}$ is proportional to the value of $\frac{C_{730} - C_{785}}{C_{730} + C_{785}}$. To demonstrate this, we plot $\lambda_{peak}$ as a function of the value of $\frac{C_{730} - C_{785}}{C_{730} + C_{785}}$ calculated by eq. (5). As illustrated in Fig. 3g, a well fitted linear curve indicates the proportional relation between $\lambda_{peak}$ and the value of $\frac{C_{730} - C_{785}}{C_{730} + C_{785}}$, in which $\frac{C_0}{k}$ and $\lambda_0$ are 34.99 and 758.2 nm, respectively. The scattering intensity is evaluated by the cross-section as

$$I_{sca} = I_{inc}\cos^2\theta C_{sca},$$ (9)

where $I_{sca}$ is scattering intensity, $I_{inc}$ is the intensity of incident polarized light, and $\theta$ is the angle between the gold nanorod longitudinal axis and the incident light polarization direction. This is why gold nanorods are reported to have direction-dependent scattering due to polarization illumination, which is a result of the anisotropic shape of the nanorods[36]. Thus, the scattering intensity-based detection methods cannot avoid the influence of the orientation dynamics of anisotropic nanoprobes during intracellular measurements.

As illustrated in Fig. 2a, we have used two parallel s-polarized lasers to eliminate interference from $\theta$. When we adjust the power of two lasers to obtain equal incident intensity, that is, $I_{inc(730)}$ is equal to $I_{inc(785)}$, $\lambda_{peak}$ can be obtained by

$$\lambda_{peak} = -\frac{C_0}{k}\frac{I_{sca(730)} - I_{sca(785)}}{I_{sca(730)} + I_{sca(785)}} + \lambda_0.$$ (10)

Eq. (10) predicts that $I_{inc(730)}$ and $I_{inc(785)}$ are the only things we need to obtain $\lambda_{peak}$ of gold nanorods. Herein, the spectral mapping is obtained by converting the two-channel images using eq. (10).

**Chemicals and general techniques.** Sodium oleate, sodium hydroxide (NaOH), copper(I) bromide (CuBr), sodium azide (NaN$_3$), copper(II) sulfate (CuSO$_4$), sodium chloride (NaCl), bis(2-hydroxyethyl) disulfide (BHEDS), 2-bromoisobutyric acid, N,N,N′,N″,N″-pentamethyldiethylenetriamine (PMDETA), N-isopropylacrylamide (NIPAAm), acrylamide (AAm), N-hydroxyethyl acrylamide (HEAAm), dicyclohexylcarbodiimide (DCC), tris(3-hydroxypropyltriazolylmethyl)amine (THPTA, 95%), and TWEEN-20 were purchased from Sigma-Aldrich (Shanghai, China). Absolute ethanol, acetone, and THF were purchased from Aladdin Reagent Inc. Alkynyl-RGD (arginine-glycine-aspartic) was purchased from GL Biochem (Shanghai) Ltd. All other reagents are of analytical grade. Ultrapure water with a resistivity of 18.2 MΩ cm was produced using a Milli-Q apparatus (Millipore) and used in the preparation of all solutions. Cover slides were purchased from Thorlabs Co., Ltd. PDMS was prepared using Sylgard 184, Dow Corning.

UV–vis spectra were recorded on a UV-1750 spectrophotometer (Shimadzu, Kyoto, Japan). High-resolution transmission electron micrographs (HRTEM) were captured on a JEOL JEM 200CX transmission electron microscope, using an accelerating voltage of 200 kV. Dynamic light scattering (DLS) experiments were performed at 25 °C using a Brookhaven BI-200SM instrument, equipped with a He-Ne laser (632.8 nm) at a fixed scattering angle of 90 °. Broad dark-field images

and spectra measurements were carried out on the same microscope as SRPS. A broadband light source (EQ-99XFC LDLS, Energetiq Technology) was used for incident illumination. True-color dark-field images are captured by a color cooled digital camera (Nikon DS-RI1), and the scattering spectra of single nanoparticles were measured by a monochromator (Acton SP2300i, PI) equipped with a spectrograph CCD (PIXIS 400BR_excelon, PI) and a grating (grating density: 300 l mm$^{-1}$; blazed wavelength: 500 nm). Fluorescent imaging was also carried out on this microscope (Ti-E, Nikon, Japan), equipped with a mercury lamp (Nikon Intensilight C-HGFI). The MTT assay was recorded at 490 nm using a microplate reader (Varioskan Flash, ThermoFisher Scientific).

**Synthesis of pNIPAAm.** The following reagents were mixed in a three neck round-bottom flask[24]: 3.68 g N-isopropylacrylamine (NIPAAm), 35 μl PMDETA, 0.015 g bis[2-(2′-bromoisobutyryloxy)ethyl]disulfide, 18 ml of deionized water, and 12 ml of methanol. The reaction solution was degassed with three cycles of freeze-pump-thaw, followed by the addition of 0.01 g CuBr. The flask was then filled with nitrogen, and the mixture was left to melt at room temperature. The reaction solution was left overnight at room temperature under mild stirring. After evaporation of the solvent, the crude product was dissolved in water and purified by dialysis to yield the pNIPAAm with LCST at 32 °C. The molecular weight of pNIPAAm was controlled by changing the amount of bis[2-(2′-bromoisobutyryloxy)ethyl]disulfide.

**Assembly and optimization of thermosensitive gold nanorods.** Gold nanorods (50 OD), with $40 \times 120$ nm dimension and 750 nm LSPR wavelength (Supplementary Fig. 8), were purchased from NanoSeedz Ltd. To assemble AuNRs@pNIPAAm, 10 μl of gold nanorods were dispersed in 20 ml of 20 mg ml$^{-1}$ pNIPAAm and then incubated at room temperature for 24 h, followed by centrifugation to remove residual pNIPAAm. To achieve the best response ability, we synthesized a series of pNIPAAm with a molecular weight from 2500 to 10,000 Da, which were then incubated with gold nanorods deposited on a cover slide surface for 12 h. We compared the mean LSPR wavelength shift of these AuNRs@pNIPAAm particles, and the greater the molecular weight of pNIPAAm, the greater the LSPR wavelength shift of the AuNRs@pNIPAAm (Supplementary Fig. 9). Finally, pNIPAAm with a 5000 Da molecular weight was selected, concerning the high viscosity of the 10,000-Da sample.

**Temperature calibration of thermosensitive gold nanorods.** To measure the accurate temperature of nanoparticles deposited on the cover slide, a thermocouple (Victor, VC86E, TP01) was dipped into the solution, touching the cover slide (Supplementary Fig. 10). Two heaters (HT10K, HT15W, Thorlabs) were attached to the objective or dipped in water. The heat power of the heaters are controlled by the thermocouple to generate a stable temperature for nanoparticles. The scattering spectra and SRPS images of the nanoparticles were both acquired at each certain temperature.

**Cell-targeting AuNRs@pNIPAAm/RGD preparation.** RGD peptides were linked to a PEG chain (HS-mPEG-N$_3$, 5000 MW) via a click reaction. First, HS-mPEG-N$_3$ (2 ml, 10 mg ml$^{-1}$) was mixed with alkynyl-RGD (500 μl, 1 mg ml$^{-1}$), which was degassed for 15 min. Subsequently, CuSO$_4$·5H$_2$O (100 μl, 7 mg ml$^{-1}$), ascorbic acid (500 μl, 7 mg ml$^{-1}$), and THPTA (100 μl, 7 mg ml$^{-1}$) were injected into the mixture. The mixed solution was stirred at room temperature for 48 h to conjugate RGD to PEG. Then, PEG-RGD was dialyzed for 48 h to remove unreacted molecules. To assemble AuNRs@pNIPAAm/RGD, gold nanorods were dispersed in the mixture of PEG-RGD and pNIPAAm. The molar ratio of pNIPAAm and PEG-RGD was characterized and optimized using UV−vis spectra and SRPS (Fig. 5).

**Cell culture and intracellular thermometry experiment.** A HeLa cell line was obtained from the Institute of Cell Biology at the Chinese Academy of Sciences (Shanghai, P. R. China) and cultured in DMEM (Life Technologies, Grand Island, NY, USA) at 37 °C under 5% CO$_2$ atmosphere, supplemented with L-glutamine (2 mM) and 10% fetal bovine serum (FBS). At the logarithmic growth phase, the cells were incubated with 40 pM AuNRs@pNIPAAm/RGD dispersed in cultured medium for 5 h. After 4 h of incubation with nanoprobes, HeLa cells were rinsed by fresh culture medium to eliminate residual free probes. To simplify the following experiment process, HeLa cell samples were treated with 4% paraformaldehyde.

**Photothermal therapy experiment.** To demonstrate the PTT performance of intracellular AuNRs@pNIPAAm/RGD at the single-particle level. HeLa cell samples were cultured with 40 pM AuNRs@pNIPAAm/RGD for 5 h before 10 min irradiation by a 795-nm laser (LE-LS-795-1100TFCA, LEO Photoelectric) with power of 3 W cm$^{-2}$. After another 19 h of incubation, cells were stained by culture media containing 2 μM calcein-AM and 4 μM propidium Iodide (PI) for 15 min, fluorescence images of stained cells were obtained in the FITC and TRITC mode, respectively. In addition, three control samples were prepared, lacking the irradiation, AuNRs@pNIPAAm/RGD incubation or both.

**Data availability.** The data and computer codes supporting the findings of this study are available from the authors upon reasonable request.

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

## Acknowledgements

This research is supported by the National Natural Science Foundation of China (Grants Nos. 21335004, 21327902, 21427807, 21605081, and 21622505), the Natural Science Foundation of Jiangsu Province (Grants No. BK20160638), China Postdoctoral Science Foundation (Grants Nos. 2016M590434, and 2017T100345). We thank Prof. Nongjian Tao for helping improve the experiments. We thank Prof. Yitao Long for helping build the broadband dark-field microscope.

## Author contributions

Z.C. and J.Z. conceived the study. X.C. and Q.X. performed the experiments. Z.C. built the microscope set-up. Z.C., X.C., Q.X. and J.-J.Z. designed the experiments. Y.C. prepared the materials. J.Z., Q.M. and H.-Y.C. advised the experiments. Z.C. analyzed the data and wrote the paper.

## Additional information

**Competing interests:** The authors declare no competing financial interests.

