## [Peer Review File · Nature Communications]

Reviewers' comments:

Reviewer #1 (Remarks to the Author):

The manuscript from Xueqin Chen et al. proposes a way of improving the current methods available for thermal imaging combined with heat generation. The idea of combining plasmonic nanoparticles with thermoresponsive polymers is quite well developed, and has been proposed in the past for SERS, drug delivery and thermometry. From the thermometry point of view, the technique was presenting some important drawbacks that would make it unpractical. Mainly, the dependence of the result on the angular position of the particles related to light polarization, plus concerns about the possibility that some other environmental properties would interfere with the thermal reading. Here, the authors propose a smart way to move forward by designing a new dark-field based imaging strategy. The paper is clear and complete, and the results could be reproduced in other laboratories if needed. With the proposed technique, the polarization problem is solved. Besides, the technique is quite fast, and this would allow for actual thermal monitoring of the particles in biological samples. However, as commented below, the application is probably restricted to studies of thermal therapy in vitro and do not include thermal mapping. Here I provide a list of comments and questions on the manuscript:

- In the introduction the authors claim that in thermometry imaging "the spatial resolution is determined by the distance between thermometer and photothermal agents". However, there are probes in which the same nanoparticle is doing both things (see for instance the works published by the group of D. Jaque on Nd³⁺-doped nanoparticles. They do thermal imaging in cells, ex vivo and in vivo). In such case, the spatial resolution is limited by the microscope resolution itself. Accordingly, the claim is not accurate.
- In terms of intracellular imaging, a nanoparticle-based strategy will never provide thermal mapping, as in order to have a full "non-pixelated" picture from inside the cell, it should have too many particles inside. This technique, though, allows control of photothermal therapy in vitro and can provide relevant information on that regard. Still, for photothermal therapy the concentration of particles inside the cell also needs to be higher, otherwise cancer cells won't die because of the thermal increase. Then, my concern here is: to what extent can the concentration of particles be increased without having neither aggregation nor toxic effects?
- Related to that: the thermometric technique is based on refractive index changes around the gold nanorods that trigger a shift of the plasmon resonance. But what would happen if two rods are close enough to interact with each other? Given the symmetric nature of pNIPAAm (Fig. 1c), aren't there rods linked to each other? Even if they are not, if the concentration of rods is increased they might be. Would this involve a limitation of the technique?
- Also related to that, carefully checking Figure 1f one sees resonances further in the near-infrared (at around 820 and 880 nm). Why? Often in the manuscript "single particle" response is mentioned. However, is it sure that this is only one particle? Please, check the effects of plasmonic coupling when two particles are close together to guarantee that this is a single particle.
- Please, revise the description of equation (2) as it seems to be describing a different equation included in the supplementary information (there's no C_0 , nor k in equation 2, instead, there's an A). Indeed, all this explanation only becomes meaningful reading the supplementary, as it uses terms such as "at this cross point" without mentioning any cross point.
- Related to that: A revision of figures 3d and 3e would also help readers. Both graphs show a different x-axis that after reading the Supplementary Information turn out to be mainly the same thing. There's no need to have two different names, then.
- About figure 5a, the authors say that the "thermal response avoids the effect from ionic strength, biomolecules adsorption and pH values". I agree on the pH part. However, I'm not so convinced about the rest. The change in the increment of LSPR when the particles are in water, in PBS or DMEM is of 2 nm or 1.5 nm (aprox). It is not a big change, but with these probes, a change of 15C is triggering a change of LSPR of max 11 nm. This means that these 2 nm are the 18% of the shift. How this actually

affects the accuracy of the measurement? Does this mean that each particle needs to be calibrated separately in order to do the experiment?

- In the discussion of results the authors briefly comment on something that is actually very important from the practical point of view. The temperature range in which these thermometers are accurate goes from 28 to 35 C. This is not only narrow, but it is also below the range of interest that should be available for biological samples. To what extent can it be modified and extended? Photothermal therapy control requires thermal readings between 35C and at least 50C, though a short therapy would actually get up to 60C. How realistic is it to achieve the needed range?

- "Scattering" is mentioned all along the manuscript. I'd say that in some cases it should be "extinction", as the measurements are not separating absorption from scattering.

- What's the length of the rods?

Reviewer #2 (Remarks to the Author):

The manuscript titled "Imaging the transient heat generation of individual nanostructures with the mechanoresponsive polymer" demonstrates an important improvement in the field of intracellular thermometry, which is of general interest to a broad audience as it has applications in photothermal therapy, drug delivery, and many other fields. The authors presented a straightforward implementation of a thermosensitive probe and showed a simple linear signal response to temperature change that is fast and relatively sensitive. I recommend accept after minor revision.

I have three comments (using the page numbering of the merged pdf file):

1. In general, the writing of this manuscript can be improved to mitigate confusion. As an example, page 6 line 111-112, I think the authors are trying to say the cross-section at 730 and at 785 nm are located on opposite sides of the peak of each curve, instead of "in a pair of symmetrical slopes".
2. Page 7 and figure 4. The LSPR wavelength of AuNRs@pNIPAAm has a large deviation -- 745 nm to 768 nm at 25oC, in fact this range is wider than the increase of LSPR wavelength of a single particle upon change of temperature from 25oC to 40oC. The authors should provide an explanation for this, or alternatively rule out the possibility that the wavelength change can be induced by factors other than temperature.
3. The authors should comment on how this technique could be applied in vivo, especially when the cells of interest are not directly exposed on the surface, would the tissue in the light path disrupt the formation of a HILO sheet?

Point-by-point response to the comments

Reviewer #1

General comment: The manuscript from Xueqin Chen et al. proposes a way of improving the current methods available for thermal imaging combined with heat generation. The idea of combining plasmonic nanoparticles with thermoresponsive polymers is quite well developed, and has been proposed in the past for SERS, drug delivery and thermometry. From the thermometry point of view, the technique was presenting some important drawbacks that would make it unpractical. Mainly, the dependence of the result on the angular position of the particles related to light polarization, plus concerns about the possibility that some other environmental properties would interfere with the thermal reading. Here, the authors propose a smart way to move forward by designing a new dark-field based imaging strategy. The paper is clear and complete, and the results could be reproduced in other laboratories if needed.

With the proposed technique, the polarization problem is solved. Besides, the technique is quite fast, and this would allow for actual thermal monitoring of the particles in biological samples. However, as commented below, the application is probably restricted to studies of thermal therapy in vitro and do not include thermal mapping. Here I provide a list of comments and questions on the manuscript:

Response: We thank the referee for reviewing our manuscript, affirming its clarity and comprehensibility, and for her/his suggestions on how to improve the manuscript. The provided comments and questions are responded point by point below:

Comment 1: In the introduction the authors claim that in thermometry imaging “the spatial resolution is determined by the distance between thermometer and photothermal agents”. However, there are probes in

which the same nanoparticle is doing both things (see for instance the works published by the group of D. Jaque on Nd³⁺-doped nanoparticles. They do thermal imaging in cells, ex vivo and in vivo). In such case, the spatial resolution is limited by the microscope resolution itself. Accordingly, the claim is not accurate.

Response: We agree that the spatial resolution is determined by the microscope itself when the thermosensitive photothermal nanoparticles is utilized, as reported in the work mentioned by the referee. Thus in our manuscript, we focus on the improvement of the spatiotemporal resolution that is vital for thermal imaging. In order to make the claim more accurate, we have replaced the mentioned sentence (page 2, line 31) “The spatial resolution is determined by the distance between thermometer and photothermal agents in consideration of the large temperature gradient, and the detection speed makes it far from the aim of imaging the transient heat generation” by the following one and added the mentioned work as reference.

“The spatial resolution could be improved by the encapsulation of photothermal agents with the thermometer¹ or the scanning-based strategy^{2, 3}, but the detection speed still makes them far from the aim of imaging the transient heat generation”

Comment 2: In terms of intracellular imaging, a nanoparticle-based strategy will never provide thermal mapping, as in order to have a full “non-pixelated” picture from inside the cell, it should have too many particles inside. This technique, though, allows control of photothermal therapy in vitro and can provide relevant information on that regard. Still, for photothermal therapy the concentration of particles inside the cell also needs to be higher, otherwise cancer cells won’t die because of the thermal increase. Then, my concern here is: to what extent can the concentration of particles be increased without having neither aggregation nor toxic effects?

Response: We agree that the nanoparticle-based strategy will never provide the full “non-pixelated” thermal mapping due to the particle density limitation in cells. Thus we preferably concern the transient heat generation at individual gold nanorods in our manuscript. High concentration of particles is necessary for the photothermal therapy. We introduce limited particles into the cell in order to make it clear for demonstrating single-nanoparticle spectra. In fact, the amount of intracellular nanoparticles is more than it looks because most of them are out of focus, which is supported by the 3D image (Supplementary Fig. 6 and Supplementary Video 4). To evaluate the extent the concentration of particles can be increased without having neither aggregation nor toxic effects, as well as the performance in PTT (see in **Methods**), we carry out the following supplementary experiments and added it to the revised manuscript (page 10, line 202).

“Commonly the photothermal agents need higher concentration to generate enough heat to kill the tumor cells, but too dense particles will induce the aggregation and cytotoxicity. Thus the concentration of intracellular AuNRs@pNIPAAm/RGD needs to be optimized. As illustrated in Supplementary Fig. 3a, aggregates start to arise in HeLa cells incubated with 40 pM AuNRs@pNIPAAm/RGD for 5 h. The aggregates have broad scattering peaks, attributed to the scattering overlapping of multiple particles (Supplementary Fig. 3b). After incubation for 8 h, massive aggregates are observed with much wider scattering peaks. Too broad scattering spectra may reduce the sensitivity. To avoid, the incubation period is controlled less than 5 h. In this condition, the AuNRs@pNIPAAm/RGD still demonstrates a good PTT performance for HeLa cells (Supplementary Fig. 4). Cytotoxicity is another essential factor on concern for foreign materials in biological system. With MTT assay, HeLa cells display no viability variation even after 48 h incubation with 80 pM of AuNRs@pNIPAAm/RGD that is twice more than the actual dosage (Supplementary Fig. 5).”

Comment 3: Related to that: the thermometric technique is based on refractive index changes around the gold nanorods that trigger a shift of the plasmon resonance. But what would happen if two rods are close enough to interact with each other? Given the symmetric nature of pNIPAAm (Fig. 1c), aren't there rods linked to each other? Even if they are not, if the concentration of rods is increased they might be. Would this involve a limitation of the technique?

Response: The coupling effect between plasmonic nanoparticles can induce a red-shift of the LSPR wavelength of each nanorod, which is well-known theoretically and observed experimentally⁴. In our manuscript, the coupling effect should induce the LSPR wavelength longer than single gold nanorods (~750 nm). Fortunately, this coupling effect is barely to observe due to the modification of pNIPAAm. The interparticle plasmonic coupling is distance-dependent and the LSPR wavelength will be affected only when the distance between gold nanorods decreases less than 10 nm. The hydrophilic pNIPAAm provides a larger hydrated radius than the bare gold nanorods (Supplementary Fig. 1) that keeps the particles away from each other in solution. Though the radius reduces at high temperature and we can observe some aggregations of gold nanorods with high density, the LSPR wavelength still keeps around 750 nm even if some spectra overlapping exist (Supplementary Fig. 3). To explain, the HRTEM image shows that the pNIPAAm layer on the gold nanorod surface has a thickness of ~6nm even in dry state (Fig. 1b), indicating that two pNIPAAm capped gold nanorods hardly get closer than 12 nm, which is longer than the required distance of the coupling effect.

The pNIPAAm has a symmetric structure, but the disulfide bond in the middle of polymer chain is the key to conjugate the pNIPAAm to the gold nanorod surface as thiolate has a strong binding towards the gold surface⁵. In the case, the gold nanorods should not be linked to each other, which is proved by the UV-vis spectra. Figure 5c displays the extinction spectra of bare gold nanorods, and gold nanorods modified by

pNIPAAm and PEG-RGD. The similar shape of spectra indicates the single-particle level gold nanorods. As mentioned above, high concentrated gold nanorods induce the aggregation but the coupling effect may not be the problem. The LSPR wavelength still keeps around 750 nm though some spectra overlapping exist (Supplementary Fig. 3b). In consideration that the spectra overlapping possibly reduces the sensitivity, the concentration of gold nanorods should be controlled at a proper level to achieve a clear single-particle scattering dots image.

Comment 4: Also related to that, carefully checking Figure 1f one sees resonances further in the near-infrared (at around 820 and 880 nm). Why? Often in the manuscript “single particle” response is mentioned. However, is it sure that this is only one particle? Please, check the effects of plasmonic coupling when two particles are close together to guarantee that this is a single particle.

Response: We’re sorry to make the referee confused, the two peaks located at 820 and 880 nm are assigned to the intrinsic spectrum of the white light source (EQ-99XFC LDLS) used for broadband dark-field imaging. The data sheet of the EQ-99XFC LDLS light is attached as a supplementary dataset (alternatively visit <http://www.energetiq.com/fiber-coupled-laser-driven-light-source-long-life-compact.php>) for detail information. The LDLS has an ultra-high brightness and the power stability are maintained across the broad spectrum except the range from 800 to 950 nm. To avoid the confusion, we have added the explanation in the revised manuscript where the scattering spectra are discussed (page 4, line 81).

In our manuscript, we claim that all the scattering spectra for thermometry are from single nanoparticle. Here we will prove it theoretically and experimentally. In theory, when two particles are close together, above all less than 10 nm, the plasmonic coupling effect would induce a remarkable red-shift of LSPR wavelength.

Otherwise the well-defined LSPR scattering peak indicates the single-particle gold nanorod. The scattering spectra displayed in Fig. 1f demonstrate the similar shape with the simulated single gold nanorod spectra (Fig. 2d), such as the half peak width and peak position. On the contrary, the aggregates of multiple gold nanorods demonstrate the wider overlapped spectra (Supplementary Fig. 3). In the thermal response experiment, the LSPR wavelength has a uniform distribution around 750 and 770 nm at 25 and 40 °C respectively (Fig. 1d, e), also indicating the single-particle status of the gold nanorods. The rotation experiment provides another evidence. No matter in solution (Fig. 3, Supplementary Video 1-2) or in cells (Supplementary Fig. 7), the scattering dots illustrate the direction-dependent scattering intensity, which is the feature of single anisotropic gold nanorod. According to the above proofs, we believe that this is a single-particle nanoprobe.

Comment 5: Please, revise the description of equation (2) as it seems to be describing a different equation included in the supplementary information (there's no C_0 , nor k in equation 2, instead, there's an A). Indeed, all this explanation only becomes meaningful reading the supplementary, as it uses terms such as "at this cross point" without mentioning any cross point.

Response: We're sorry for making this error in the description of equation (2). We wanted to avoid the complex formula derivation in the results section and use the constant A instead of $-C_0/k$ in equation (9). To correct it, we have replaced the description by the following sentence (page 6, line 126).

"where A and λ_0 are constant, I_{sca} is scattering intensity of each channel."

Comment 6: Related to that: A revision of figures 3d and 3e would also help readers. Both graphs show a different x-axis that after reading the Supplementary Information turn out to be mainly the same thing. There's

no need to have two different names, then.

Response: We have changed the x-axis of Fig. 3e to make them have same name.

Comment 7: About figure 5a, the authors say that the “thermal response avoids the effect from ionic strength, biomolecules adsorption and pH values”. I agree on the pH part. However, I’m not so convinced about the rest. The change in the increment of LSPR when the particles are in water, in PBS or DMEM is of 2 nm or 1.5 nm (aprox). It is not a big change, but with these probes, a change of 15C is triggering a change of LSPR of max 11 nm. This means that these 2 nm are the 18% of the shift. How this actually affects the accuracy of the measurement? Does this mean that each particle needs to be calibrated separately in order to do the experiment?

Response: We agree that the description of Fig. 5a is not precise. Actually we intended to claim that the probes keep the thermal response in various solution, but the difference behavior in aqueous, PBS and DMEM should not be neglected. To improve the expression, we have replaced the description for Fig. 5a “thermal response avoids the effect from ionic strength, biomolecules adsorption and pH values” (page 9, line 186) by the following sentence.

“The AuNRs@pNIPAAm keeps the thermometry ability in broad pH range solutions and culture medium though the ionic strength and biomolecules still affect the sensitivity”

To avoid the effect from ionic strength and biomolecules to the thermometry accuracy, we propose two ways. The first one is the calibration for each probe, as the referee mentioned. This strategy is accurate and suitable for studying the rare nanoparticle samples, as shown in Fig. 4d, e. When the probes are used to image

the intracellular temperature, the calibration for each probe is inefficient. Thus we use an easier process instead, which is described in line 225 of page 11. Simply, the probes show a LSPR peak change as a function of applied 808 nm laser power (Fig. 6c). It is similar to the temperature-dependent LSPR peak curves (Fig.4b), that is, the linear range from 762 to 769 nm indicates the temperature range from 30 to 35 °C. Thus the temperature is easily measured in this range without any calibration. The result in Fig. 6d is obtained by this way.

Comment 8: In the discussion of results the authors briefly comment on something that is actually very important from the practical point of view. The temperature range in which these thermometers are accurate goes from 28 to 35 C. This is not only narrow, but it is also below the range of interest that should be available for biological samples. To what extent can it be modified and extended? Photothermal therapy control requires thermal readings between 35C and at least 50C, though a short therapy would actually get up to 60C. How realistic is it to achieve the needed range?

Response: As the referee mentioned, the thermometry range from 28 to 35 °C is not only narrow but also below the required temperature for photothermal therapy. However, the mentioned drawbacks could be improved by some modification of the thermometer preparation. The temperature detection range is determined by the modified thermosensitive polymer layer on the gold nanorod surface. In our manuscript, we use the pure pNIPAAm whose low critical solution temperature (LCST) is ~32 °C in consideration of simplifying the experiment model. To extend this range, integration of acrylamide (AAm) into the pNIPAAm allows for tuning of the LCST across a wide range of temperatures⁶. According to the literature, the LCST of copolymer pNIPAAm-co-pAAm can be tuned up to 55 °C when the ratio of NIPAAm and AAm is 3:1, which is

suitable for the photothermal therapy^{5,6}.

To demonstrate the range that can be extended in our technique, we have performed and added the following supplementary experiment to the revised manuscript at page 5, line 89.

“This thermometry range could be extended by the integration of acrylamide (AAm) into the pNIPAAm in consideration that photothermal therapy control requires thermal readings between 35 °C and at least 50 °C. As shown in Supplementary Fig. 2, the LSPR wavelength of gold nanorods modified by the copolymer pNIPAAm-co-pAAm with LCST at 39 °C illustrate a two-stage s-shape curve over the temperature. Two linear ranges locate at 30-35 °C and 42-48.5 °C respectively. The first linear range reproduces the pure pNIPAAm capped gold nanorods, while the second one is attributed to the integration of AAm. The extended thermometry range offers the application potential in photothermal therapy control.”

Comment 9: “Scattering” is mentioned all along the manuscript. I’d say that in some cases it should be “extinction”, as the measurements are not separating absorption from scattering.

Response: We’re sorry for making the referee confused about our optical setup. All the spectra (both the broadband and two-channel ones) in our manuscript are obtained by the dark-field microscope except the UV-vis extinction spectra shown in Fig. 5c and Supplementary Fig. 8b. The dark-field microscope setup is illustrated in Fig. 2a, where the incident light is not collected by the objective and only the scattering light is directed to the camera^{7,8}. Thus in most case, we are not able to get the absorption feature of the nanoparticles using the dark-field microscope, which is commonly transferred to the thermal dissipation^{9,10}. For simulations, we also only calculated the scattering cross-section of nanoparticles.

Comment 10: What's the length of the rods?

Response: According to the statistic result from HRTEM images, the radius and length of rods are estimated to be 40 and 120 nm respectively. This data has been added to the description of Supplementary Fig. 8 in the revised manuscript (page 16, line 350).

Reviewer #2

General comment: The manuscript titled "Imaging the transient heat generation of individual nanostructures with the mechanoresponsive polymer" demonstrates an important improvement in the field of intracellular thermometry, which is of general interest to a broad audience as it has applications in photothermal therapy, drug delivery, and many other fields. The authors presented a straightforward implementation of a thermosensitive probe and showed a simple linear signal response to temperature change that is fast and relatively sensitive. I recommend accept after minor revision.

I have three comments (using the page numbering of the merged pdf file):

Response: We thank the referee for reviewing our manuscript, affirming its clarity and comprehensibility, and for her/his suggestions on how to improve the manuscript. The three comments are responded point by point below:

Comment 1: In general, the writing of this manuscript can be improved to mitigate confusion. As an example, page 6 line 111-112, I think the authors are trying to say the cross-section at 730 and at 785 nm are located on opposite sides of the peak of each curve, instead of "in a pair of symmetrical slopes".

Response: To mitigate the confusion, we have replaced the sentence “located in a pair of symmetrical slopes” by “located on opposite sides of the peak of each curve” in the revised manuscript (page 6, line 120).

Comment 2: Page 7 and figure 4. The LSPR wavelength of AuNRs@pNIPAAm has a large deviation -- 745 nm to 768 nm at 25oC, in fact this range is wider than the increase of LSPR wavelength of a single particle upon change of temperature from 25oC to 40oC. The authors should provide an explanation for this, or alternatively rule out the possibility that the wavelength change can be induced by factors other than temperature.

Response: The large deviation of LSPR peaks comes from different gold nanorod particles in the same sample. However, these particles demonstrate comparable LSPR peak shifts to temperature increase (Fig. 1g and Supplementary Fig. 2). This peak shift is what we focus in our manuscript. To rule out the wavelength change induced by factors other than temperature, we investigate the behavior of the same AuNRs@pNIPAAm particles in different environment, including water, PBS and culture medium DMEM (Fig. 5a). The AuNRs@pNIPAAm particles keep comparable peak shift upon change of temperature from 25 °C to 40 °C in various solution. To eliminate the deviation from different nanoparticles, we propose two different ways to achieve the accurate thermometry. The first one is the calibration for each probe. This strategy is accurate and suitable for studying the rare nanoparticle samples, as shown in Fig. 4d, e. When the probe is used to image the intracellular temperature, the calibration for each probe is inefficient. Thus we use an easier process instead, which is described in line 225 of page 11. Simply, the probes show a LSPR peak change as a function of applied 808 nm laser power (Fig. 6c), fitting the temperature-dependent LSPR peak curves (Fig.4b). Thus the linear range from 762 to 769 nm indicates the temperature range from 30 to 35 °C. Thus the temperature is easily measured in this range without any calibration. The result in Fig. 6d is obtained by this way.

Comment 3: The authors should comment on how this technique could be applied *in vivo*, especially when the cells of interest are not directly exposed on the surface, would the tissue in the light path disrupt the formation of a HILO sheet?

Response: We thank the referee's advice about the potential application *in vivo*. The HILO sheet technique based on TIRF objective is hard to be used to illuminate the deep tissue. However the most essential part of our technique is not the HILO sheet illumination but the spatiotemporal resolution plasmonic spectroscopy based on two-channel dark-field imaging, which has the application potential *in vivo*. For example, the endoscope system has been widely used *in vitro* and *in vivo*, which can be integrated with a lot of techniques, such as electrochemical sensor¹¹, fluorescent probing¹²⁻¹⁴ and optogenetics¹⁵. An endoscope system integrated with our two-channel laser illumination technique may allow us to monitor the plasmonic nanoprobe in deep tissue or cells. We have added the following comment to the discussion section in the revised manuscript (page 12, line 252).

“the endoscope system equipped with the SRPS technique may allow for monitoring the plasmonic nanoprobe in deep tissue or cells, which will expand the potential application *in vivo*”.

References

1. Rocha, U., *et al.* Subtissue Imaging and Thermal Monitoring of Gold Nanorods through Joint Encapsulation with Nd-Doped Infrared-Emitting Nanoparticles. *Small* **12**, 5394-5400 (2016).
2. Rodríguez-Sevilla, P., *et al.* Thermal Scanning at the Cellular Level by an Optically Trapped Upconverting Fluorescent Particle. *Adv Mater* **28**, 2421-2426 (2016).

3. Chen, Z., *et al.* Imaging Local Heating and Thermal Diffusion of Nanomaterials with Plasmonic Thermal Microscopy. *ACS nano* **9**, 11574–11581 (2015).
4. Sonnichsen, C., Reinhard, B. M., Liphardt, J., Alivisatos, A. P. A molecular ruler based on plasmon coupling of single gold and silver nanoparticles. *Nat Biotech* **23**, 741-745 (2005).
5. Yavuz, M. S., *et al.* Gold nanocages covered by smart polymers for controlled release with near-infrared light. *Nat Mater* **8**, 935-939 (2009).
6. Barhoumi, A., *et al.* Photothermally Targeted Thermosensitive Polymer-Masked Nanoparticles. *Nano Lett* **14**, 3697-3701 (2014).
7. Anker, J. N., *et al.* Biosensing with plasmonic nanosensors. *Nat Mater* **7**, 442-453 (2008).
8. Liu, G. L., *et al.* Quantized plasmon quenching dips nanospectroscopy via plasmon resonance energy transfer. *Nat Methods* **4**, 1015-1017 (2007).
9. Clavero, C. Plasmon-induced hot-electron generation at nanoparticle/metal-oxide interfaces for photovoltaic and photocatalytic devices. *Nat Photon* **8**, 95-103 (2014).
10. Brongersma, M. L., Halas, N. J., Nordlander, P. Plasmon-induced hot carrier science and technology. *Nat Nanotechnol* **10**, 25-34 (2015).
11. Pan, R. R., *et al.* Nanokit for single-cell electrochemical analyses. *P Natl Acad Sci USA* **113**, 11436-11440 (2016).
12. Lee, H., *et al.* An endoscope with integrated transparent bioelectronics and theranostic nanoparticles for colon cancer treatment. *Nat Commun* **6**, (2015).
13. Yan, R., *et al.* Nanowire-based single-cell endoscopy. *Nat Nano* **7**, 191-196 (2012).
14. Singhal, R., *et al.* Multifunctional carbon-nanotube cellular endoscopes. *Nat Nano* **6**, 57-64 (2011).
15. Reutsky-Gefen, I., *et al.* Holographic optogenetic stimulation of patterned neuronal activity for vision restoration. *Nat Commun* **4**, (2013).

REVIEWERS' COMMENTS:

Reviewer #1 (Remarks to the Author):

The authors did a great job to answer all the comments. In my opinion, the manuscript is now ready for publication.

Reviewer #2 (Remarks to the Author):

This revised version of an earlier submission has responded in a satisfactory way to most of the criticisms raised. I still think the writing needs significant work before this manuscript can be published.

Point-by-point response to the comments

Reviewer #1

The authors did a great job to answer all the comments. In my opinion, the manuscript is now ready for publication.

Response: We thank the referee for reviewing our manuscript, affirming its clarity and comprehensibility, and for her/his suggestions on how to improve the manuscript.

Reviewer #2

This revised version of an earlier submission has responded in a satisfactory way to most of the criticisms raised. I still think the writing needs significant work before this manuscript can be published.

Response: We thank the referee for reviewing our manuscript, affirming its clarity and comprehensibility, and for her/his suggestions on how to improve the manuscript. The writing has been improved by Nature Research Editing Service.